# Bellman Optimal Stepsize Straightening of Flow-Matching Models

**Bao Nguyen**
VinUniversity
`bao.nn2@vinuni.edu.vn`

**Binh Nguyen**
National University of Singapore
`binhnt@nus.edu.sg`

**Viet Anh Nguyen**
Chinese University of Hong Kong
`nguyen@se.cuhk.edu.hk`

## Abstract

Flow matching is a powerful framework for generating high-quality samples in various applications, especially image synthesis. However, the intensive computational demands of these models, especially during the finetuning process and sampling processes, pose significant challenges for low-resource scenarios. This paper introduces Bellman Optimal Stepsize Straightening (BOSS) technique for distilling flow-matching generative models: it aims specifically for a few-step efficient image sampling while adhering to a computational budget constraint. First, this technique involves a dynamic programming algorithm that optimizes the stepsizes of the pretrained network. Then, it refines the velocity network to match the optimal step sizes, aiming to straighten the generation paths. Extensive experimental evaluations across image generation tasks demonstrate the efficacy of BOSS in terms of both resource utilization and image quality. Our results reveal that BOSS achieves substantial gains in efficiency while maintaining competitive sample quality, effectively bridging the gap between low-resource constraints and the demanding requirements of flow-matching generative models. Our paper also fortifies the responsible development of artificial intelligence, offering a more sustainable generative model that reduces computational costs and environmental footprints. Our code can be found at https://github.com/nguyenngocbaocmt02/BOSS.

## 1 Introduction

There have been impressive advancements in deep generative models in recent years, which constitute an appealing set of approaches capable of approximating data distributions and generating high-quality samples, as showcased in influential works such as Ramesh et al. (2022); Saharia et al. (2022); Rombach et al. (2022). They are primarily driven by a category of time-dependent generative models that utilize a predefined probability path, denoted as $\{\pi_t\}_{t\in[0,1]}$. This probability path is a process that interpolates between the initial noise distribution $\pi_0$ and the target data distribution $\pi_1$. The training for these models can be broadly characterized as a regression task involving a neural network function $v_\theta$ and a target ideal velocity $v_t(x)$:

$$\mathcal{L}(\theta) := \mathbb{E}_{t\in[0,1],\ X_t\sim\pi_t}[\ell(v_\theta(X_t,t),v_t(X_t))].$$

Here, the velocity network $v_\theta$ maps any input data $x$ at time $t \in [0,1)$ to a vector-valued velocity quantity $v_\theta(x,t)$ and plays a crucial role in the generation of samples from the interpolation process through the relationship:

$$X_1 = X_0 + \int_0^1 v_\theta(X_t,t)\mathrm{d}t,$$

which is the solution of the ODE $\mathrm{d}X_t = v_\theta(x_t,t)\mathrm{d}t$ with the boundary condition $X_{t=0} = X_0$. A noteworthy class of algorithms that fits within this framework includes denoising diffusion models (Ho et al., 2020; Sohl-Dickstein et al., 2015; Song et al., 2020b) and the more recent flow-

matching/rectified-flow models (Liu et al., 2022b; Lipman et al., 2022; Albergo & Vanden-Eijnden, 2022; Neklyudov et al., 2023).

The latter type of model extends the principles employed in training diffusion models to simulation-free continuous normalizing flows (CNF, Chen et al. 2018). It is particularly attractive because it fixes the suboptimal alignment between noises and images of diffusion models by introducing a straight trajectory formula connecting them. This leads to (empirically observed) faster training and inference time than diffusion models. The rectified flow framework Liu et al. (2022b) also includes a technique called *reflow*, which gradually rectifies the probability paths. It significantly reduces the number of function evaluations needed for sampling and thus belongs to the family of distillation methods.

However, the standard reflow technique proposed in Liu et al. (2022b) requires a significant amount of computational budget: on a small-dimensional dataset such as CIFAR-10 ($32 \times 32$ pixel images), it takes at least an additional 300,000 *re*training iterations of the pretrained velocity networks to reach FID (Fréchet Inception Distance, Heusel et al. 2017) of 4.85 for 1-step generation. The additional retraining time can reach approximately 200 days of A100 GPU for distilling models for 1-step sampling on higher-dimensional scale datasets to achieve competitive FID, as stated in an extension of the rectified flow framework (Liu et al., 2023). Motivated to fix this problem, in this work, we aim to distill the Rectified Flow model while satisfying the following objective.

> Given a pretrained flow-matching velocity $v_\theta$ and a target of $K$ number of function evaluations (NFEs), how can we adapt $v_\theta$ with a proper sampling schedule to generate high-fidelity images using only a modest computational resource?

**Contributions.** We propose BOSS, the Bellman Optimal Stepsize Straightening method, to finetune pretrained flow-matching models. Our proposal includes two phases. The first phase seeks the optimal $K$-element sequence $\Delta^*$ for the initial model $v_\theta$. The second phase utilizes $\Delta^*$ to retrain $v_\theta$ such that the retrained model $v_{\theta^*}$ performs better. With the proposed procedure, we straighten the velocity network with just about 10,000 retraining iterations while outperforming the standard reflow strategies regarding image quality. Quantitatively, our procedure consistently achieves lower FID in unconditional image generation with four different datasets. Furthermore, as the additional results in Appendix E show, the straightening procedure using Low-Rank Adaptation (LoRA) can finetune only $2\%$ of the model's parameters, yet it performs competitively to that of full-rank finetuning.

**Related works.** Our approach is directly related to existing works on improving the sampling efficiency of diffusion and flow-matching models with *training-based algorithms*. In Salimans & Ho (2021), the authors proposed an approach to enhance the sampling speed of unguided diffusion models through iterative distillation. This is extended to the case of classifier-free guided diffusion models in Meng et al. (2023). In Wang et al. (2023), the authors propose a method leveraging reinforcement learning to automatically search for an optimal sampling schedule for Diffusion Probabilistic Models (DPMs), addressing limitations in hand-crafted schedules and the assumption of uniformity across instances. Our work has a few common features and motivations with Watson et al. (2021), which achieved significant speed-ups through dynamic programming and decomposed loss terms. However, their focus on individual Kullback–Leibler divergence loss that neglects the cumulative information loss during sampling. This results in images with reduced overall quality. In contrast, we focus on minimizing the local truncation error during the sampling procedure, which improves the image quality consistently across all budgets of NFEs. Moreover, we propose a finetuning method that allows faster sampling with just a few NFEs. Recent work by Song et al. (2023) introduced a framework that learns a model capable of mapping any point at any time to the trajectory's starting point, called the *consistency model*. After submitting this work, we discovered a concurrent study by Li et al. (2023). This work pointed out that using uniform stepsizes is suboptimal for diffusion model sampling and instead used evolutionary algorithms to search for the optimal stepsizes and score network architectures, with the FID score being the optimized metric.

Within the context of Rectifed Flow/Flow Matching, Liu et al. (2022b) proposed a reflow method that uses retraining to straighten the probability sampling path. This results in a low NFE sampling with favorable image quality. The recent work of Liu et al. (2023) takes this framework to a larger scale, demonstrating impressive results on high-resolution image datasets. However, both rely on computational intensive retraining procedures, which we improve in our work.

The other direction that aims to accelerate the sampling process of diffusion/flow matching models is *training-free samplers* (Song et al., 2020a; Bao et al., 2022; Liu et al., 2022a; Tachibana et al., 2021; Zhang & Chen, 2022; Karras et al., 2022; Lu et al., 2022; Zheng et al., 2023). Although required no additional training step, these works mainly relied on the properties of the SDE/probability flow ODE dynamics to propose heuristic solvers/diffusion noise schedulers. Therefore, verifying whether the proposed sampling stepsizes are optimal is hard.

## 2 BACKGROUND

Suppose we are given a (pretrained) model $v_\theta(X_t, t)$ with parameter $\theta$, which is an estimator of the function $v$ from an ordinary differentiable equation (ODE) on the span $t \in [0, 1]$:

$$\mathrm{d}X_t = v(X_t, t)\mathrm{d}t. \tag{1}$$

In generative modeling with diffusion/flow matching models, this dynamic system is called the *probability flow ODE* (Song et al., 2020b; Lipman et al., 2022). The estimator $v_\theta$ allows us to flow from the distribution $\pi_0$ (noises) to the distribution $\pi_1$ (real images) via the equation:

$$X_1 = X_0 + \int_0^1 v_\theta(X_t, t)\mathrm{d}t, \tag{2}$$

where $X_0 \sim \pi_0$ and $X_1 \sim \pi_1$. In the context of our problem, $X_0$ is observable. $X_1$ is only determined by the equation (2) which is a deterministic process that for each $X_0 = x_0$, there is only value $X_1 = x_1$ coupling with it through the following equation:

$$x_1 = x_0 + \int_0^1 v_\theta(x_t, t)\mathrm{d}t.$$

We are interested in the low-cost estimate of the integral $\int_0^1 v_\theta(x_t, t)\mathrm{d}t$ with respect to $t$ over the interval $[0, 1]$. It is an essential concern when calculating the velocity field is computationally expensive. The amount of times calling $v_\theta$ is defined as the number of function evaluations (NFE).

**First Order Sampling scheme.** To solve for the integration that appears in the sampling equation (2), it is necessary to invoke a numerical integrator that uses discretized time steps. Any numerical integration scheme will induce truncation errors, which can be quantified in two forms: first, when we have the value at the previous time step $X_{\tau-\delta}$, the solver estimates the subsequent true value $X_\tau$ as $\hat{X}_\tau$, causing a *local truncation error* $X_\tau - \hat{X}_\tau$. These local errors accumulate over the number of intervals, eventually resulting in a cumulative error known as the *global truncation error*. The most popular discretization scheme is perhaps Euler's method: given a budget $K$ number of function evaluations (NFEs), the Euler uniform sampling computes the interval $\Delta = 1/K$, and the sample successively

$$x_{k/K}^i = x_{(k-1)/K}^i + v_\theta(x_{(k-1)/K}^i, (k-1)\Delta) \times \Delta \quad \forall k = 1, \dots, K, \tag{3}$$

with the initial condition $x_0^i \sim \pi_0$. We denote this uniform sampling scheme by $\mathcal{E}^U(K)$. Euler's method with uniform stepsizes $\Delta$ has local truncation error $O(\Delta^2)$, and global truncation error $O(\Delta)$. One can generalize the Euler sampling with *non-uniform* intervals by dividing the time domain $[0, 1]$ into unequal intervals with timestamps $0 = \tau_0 < \tau_1 \dots < \tau_K = 1$, and sample successively

$$x_{\tau_k}^i = x_{\tau_{k-1}}^i + v_\theta(x_{\tau_{k-1}}^i, \tau_{k-1}) \times (\tau_k - \tau_{k-1}) \quad \forall k = 1, \dots, K, \tag{4}$$

with the initial condition $x_0^i \sim \pi_0$. The timestamps equivalently determine the stepsize $\tau_k - \tau_{k-1}$ for each sampling iteration. We denote this scheme by $\mathcal{E}(\{\tau_0, \tau_1, \dots, \tau_K\})$. If the timestamps $\tau_k$ are equally spaced in $[0, 1]$, then we obtain the equivalence $\mathcal{E}(\{\tau_0, \tau_1, \dots, \tau_K\}) \equiv \mathcal{E}^U(K)$. Since the reflow procedure in Liu et al. (2022b) deals exclusively with Euler's method for being the fastest with a fixed computational budget, we focus only on this method in our paper.

## 3 OPTIMAL SAMPLING STEPSIZES

The objective presented in Section 1 can be defined more rigorously as follows. Given a pretrained model $v_\theta$ and an insignificant value of $K$, we aim to find the optimal value $\theta^*$ and sequence $\Delta^*$ such

that $\mathcal{E}(., \Delta^*)$ is a reasonable estimate for the coupling sample $x_1$ of any $x_0$. This can be posed as an integer optimization problem of finding the best schedule for sampling. Given a fixed budget of $K$ NFEs, we find a schedule $\{\tau_0, \tau_1, \ldots, \tau_K\}$ satisfying $0 = \tau_0 < \tau_1 < \ldots < \tau_K = 1$ and that the associated Euler non-uniform sampling scheme $\mathcal{E}(\{\tau_0, \tau_1, \ldots, \tau_K\})$ has minimal sampling error for the pretrained velocity $v_\theta$. In Section 3.1, we describe our estimate of the sampling error for any valid schedule. Section 3.2 presents an integer programming formulation to find the optimal stepsizes for sampling, and Section 3.3 provides a dynamic programming algorithm to find the optimal schedule.

## 3.1 Sampling Error Estimation

Given any two arbitrary timestamps $0 \leq t_j < t_k \leq 1$, we are interested in estimating the *local* Euler truncation error, i.e., measuring the discrepancy between the true value

$$X_{t_k} = X_{t_j} + \int_{t_j}^{t_k} v_\theta(X_t, t)\mathrm{d}t$$

and the one-step Euler sampling value

$$X_{t_k}^{\mathcal{E}} = X_{t_j} + v_\theta(X_{t_j}, t_j) \times (t_k - t_j),$$

where $X_{t_j}$ is sampled from the distribution $\pi_{t_j}$ which is induced by the initial distribution $\pi_0$ of $X_0$ and the ODE (1). This local truncation error can be formalized as

$$c_{jk}^{\text{truncation}} := \mathbb{E}_{X_{t_j} \sim \pi_{t_j}} \left[ \left\| \int_{t_j}^{t_k} v_\theta(X_t, t)\mathrm{d}t - v_\theta(X_{t_j}, t_j)(t_k - t_j) \right\|_2^2 \right].$$

Unfortunately, computing $c_{jk}^{\text{truncation}}$ is computationally intensive because of both the expectation operator and the integration. We instead employ the following two simplifications:

1. We fix the possible choice of time-stamps: for a sufficiently large number $K^{\max}$, the anchoring timestamps are $\{t_k\}_{k=0,\ldots,K^{\max}}$ with $t_k = k/K^{\max}$. In doing so, we have restrained the space of all possible sampling schedules to the combinations of finite anchoring timestamps $\{t_k\}$. Later in the numerical experiments, we choose $K^{\max} = 100$, leading to the anchoring timestamps $\{0, 0.01, 0.02, \ldots, 0.99, 1\}$.

2. We approximate the local truncation error $c_{jk}^{\text{truncation}}$ for any two anchoring timestamps $t_j < t_k$ by a sample average estimator $c_{jk}$ that is constructed as follows:

$$c_{jk} = \frac{1}{N} \sum_{i=1}^{N} c_{jk}^i, \quad \text{where} \quad c_{jk}^i = \|x_{t_k}^i - x_{t_j}^i - v_\theta(x_{t_j}^i, t_j) \times (t_k - t_j)\|_2^2, \tag{5}$$

where for sample $i$, the noise $x_0^i$ is drawn from $\pi_0$, and $x_{t_j}^i$ and $x_{t_k}^i$ are extracted from Euler uniform sampling path starting from $x_0^i$, taken at time $t_j$ and $t_k$, respectively.

Figure 1 illustrates how the sampling error $c_{jk}^i$ are calculated for the simple case with $K^{\max} = 5$ (or equivalently, with a uniform interval $\Delta = 0.2$). First, noise $x_0^i$ is drawn from $\pi_0$, and the blue curve depicts the nonlinear trajectory following the ODE (1). The piecewise linear trajectory is the path generated by the uniform Euler sampling with $K^{\max}$ NFEs, leading to the observed trajectory $\{x_k^i\}_{k=0,\ldots,K^{\max}}$. For a concrete example of computing $c_{25}^i$, we measure the difference between the value of a one-step Euler sampling from $t_2$ to $t_5$ with a stepsize $t_5 - t_2 = 3\Delta$ to obtain $x_2^i + v_\theta(x_2^i, t_2) \times 3\Delta$, and the observed value $x_5^i$. Intuitively, we can view $c_{jk}$ as the difference between the Euler one-step and the Euler $(k - j)$-step uniform sampling between $t_j$ and $t_k$.

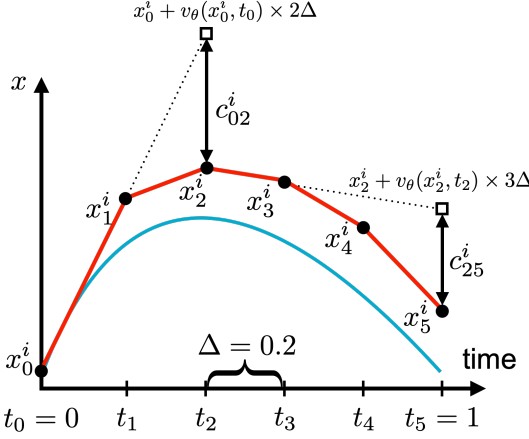

Figure 1: An example with $K^{\max} = 5$ to illustrate the computation of the sampling error.

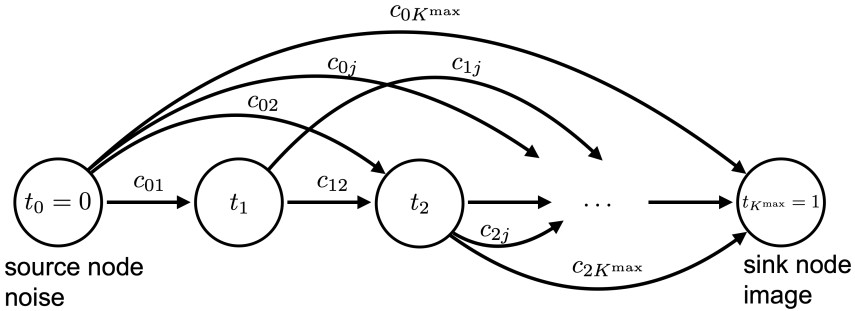

Figure 2: A network flow formulation to find the optimal sampling schedule for image generation. Time $t_0 = 0$ represents noise, while $t_{K^{\max}} = 1$ is the terminal data (images). Each discretized timestamp is represented by a node, with edges reflecting the one-dimensional flow of time from noise to image. The cost $c_{jk}$ associated with each edge is the sampling error estimate, measured by the average difference between the Euler one-step and the Euler $(k - j)$-step sampling between $t_j$ and $t_k$, see Section 3.1.

One may observe that $c_{jk}$ is only an approximation of the true local truncation error $c_{jk}^{\text{truncation}}$ because $c_{jk}$ is computed based on the Euler trajectory (red piecewise linear path in Figure 1), while the truncation error $c_{jk}^{\text{truncation}}$ should be computed based on the nonlinear trajectory of the ODE (blue curve in Figure 1). One downside of using $c_{jk}$ is that for any two consecutive timestamps $t_j$ and $t_{j+1}$, we have $c_{j,(j+1)} = 0$. This downside can be mitigated by taking $K^{\max}$ sufficiently large. On the other hand, as $K^{\max}$ gets large, calculating all the values $c_{jk}$ is computationally intensive because there are, in total, $K^{\max}(K^{\max} - 1)/2$ pair of timestamps whose errors are to be computed. Nevertheless, we demonstrate empirically in Section 5 that even when $c_{jk}$ is computed using a small number $N$ of samples, the resulting optimal schedule already demonstrates a superior performance vis-à-vis competing methods.

## 3.2 Integer Programming Formulation

As we now describe, finding the optimal sampling schedule can be formulated as a network-flow-based problem (Ahuja et al., 1993). First, construct a graph of $K^{\max} + 1$ nodes; each node represents one timestamp, see Figure 2. There is an edge connecting node $t_j$ to node $t_k$ if $t_j < t_k$, and this edge is associated with a sampling error cost $c_{jk}$, computed in Section 3.1. For a target of $K$ NFEs, the optimal sampling schedule is a path that traverses from the source node $t_0$ to the sink node $t_{K^{\max}}$ that is comprised of exactly $K$ edges. This path can be recovered from the optimal solution of the problem

$$
\begin{aligned}
\min \quad & \sum_{j=0}^{K^{\max}-1} \sum_{k=j+1}^{K^{\max}} c_{jk} z_{jk} \\
\text{s.t.} \quad & \sum_{j=0}^{K^{\max}-1} \sum_{k=j+1}^{K^{\max}} z_{jk} = K \\
& \sum_{k=1}^{K^{\max}} z_{0k} = 1, \quad \sum_{j=0}^{K^{\max}-1} z_{jK^{\max}} = 1 \\
& \sum_{k=0}^{j-1} z_{kj} = \sum_{k=j+1}^{K^{\max}} z_{jk} \qquad & \forall j \in [\![1, K^{\max-1}]\!] \\
& z_{jk} \in \{0, 1\} \qquad & \forall 0 \le j < k \le K^{\max}.
\end{aligned}
\tag{6}
$$

Above, $z_{jk} \in \{0, 1\}$ is a binary decision variable, $z_{jk} = 1$ if the path takes a one-step sampling from time $t_j$ to time $t_k$. The objective function of (6) minimizes the path's accumulated sampling error, which approximates the *global* truncation error of the Euler sampling with the corresponding step sizes. The first constraint indicates that the path should consist of exactly $K$ edges; the second constraint imposes that $t_0$ and $t_{K^{\max}}$ are the source and sink nodes, respectively. Finally, the last set of constraints is the flow conservation on each intermediary node between $t_0$ and $t_{K^{\max}}$.

### 3.3 DYNAMIC PROGRAMMING ALGORITHM

While the integer programming problem (6) can be solved using commercial solvers such as GUROBI or using network flow algorithms (see Skiena (2008, §6) for an example), there are practical cases in which we need to find optimal paths for *multiple* values of the budget $K$ NFEs. A convenient way to address this computation is to leverage a dynamic programming formulation, which successively builds up the error-to-go function at each node and for each number of remaining NFEs. To this end, for any timestamp $t_{\widehat{j}}$ and any number of remaining NFEs $\widehat{k} \in [\![1, K^{\max}]\!]$, we define the error-to-go function as

$$
V(\widehat{j}, \widehat{k}) := \begin{cases}
\min & \sum_{j=\widehat{j}}^{K^{\max}-1} \sum_{k=j+1}^{K^{\max}} c_{jk} z_{jk} \\
\text{s.t.} & \sum_{j=\widehat{j}}^{K^{\max}-1} \sum_{k=j+1}^{K^{\max}} z_{jk} = \widehat{k} \\
& \sum_{k=\widehat{j}+1}^{K^{\max}} z_{\widehat{j}k} = 1, \quad \sum_{j=\widehat{j}}^{K^{\max}-1} z_{jK^{\max}} = 1 \\
& \sum_{k=\widehat{j}}^{j-1} z_{kj} = \sum_{k=j+1}^{K^{\max}} z_{jk} & \forall j \in [\![\widehat{j}+1, K^{\max-1}]\!] \\
& z_{jk} \in \{0, 1\} & \forall \widehat{j} \le j < k \le K^{\max}.
\end{cases}
\tag{7}
$$

The error-to-go $V(\widehat{j}, \widehat{k})$ is the minimal sampling error accumulated from time $t_{\widehat{j}}$ to the terminal time $t_{K^{\max}} = 1$ using exactly $\widehat{k}$ NFEs. It is easy to see that the optimal value of problem (6) is equal to $V(0, K)$. At initialization, we set the base case

$$
V(\widehat{j}, 1) = c_{\widehat{j}K^{\max}} \quad \forall \widehat{j} \in [\![1, K^{\max}]\!].
$$

The dynamic programming update step rolls backward following

$$
\forall k = 2, \ldots, K^{\max} : \qquad V(j, k) = \min_{j < \widehat{j} \le K^{\max}} c_{j\widehat{j}} + V(\widehat{j}, k-1).
\tag{8}
$$

The output of the dynamic programming algorithm is the error function $V$, and one can assess the Bellman optimal schedule by tracing the minimizing path following (8) for each value $K$ of NFEs.

## 4 STRAIGHTENING FLOWS WITH BELLMAN STEPSIZE

Given the Bellman optimal stepsizes, we describe a piecewise linear straightening of the sampling curve. The straightening procedure aims to re-align the velocity network $v_\theta$ to reduce the accumulated sampling error at the terminal timestamps. For a fixed number of NFEs $K$, let $\{\tau_0, \ldots, \tau_K\}$ be the optimal timestamps found in Section 3 with $\tau_0 = 0$ and $\tau_K = 1$, which corresponds to $K$ stepsizes defined by $\tau_k - \tau_{k-1}$ for $k = 1, \ldots, K$. We now modify the network weights to straighten the sampling path on each interval $[\tau_k, \tau_{k+1}]$. An intuitive explanation for the straightening procedure is illustrated in Figure 3: here, suppose that $K = 2$, and the optimal schedule is $\{\tau_0 = 0, \tau_1 = 0.4, \tau_2 = 1\}$. For the sample $i$ drawn in Figure 3, the Bellman sampling induces a two-piece linear path $x_0^i \to x_2^i \to x_5^i$ (dashed line). If the velocity vectors evaluated at $x_0^i$ and $x_2^i$ align with the dashed line, then the Bellman optimal Euler sampling with $K = 2$ incurs zero loss compared to the Euler uniform

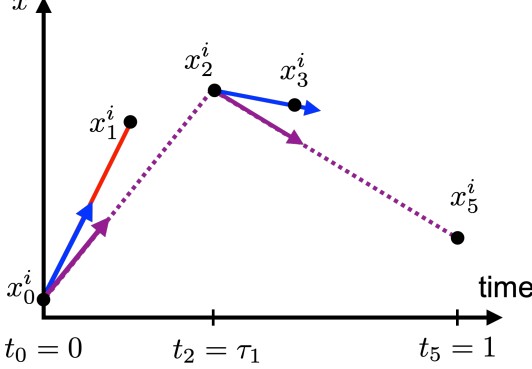

Figure 3: Continued example following Figure 1 for straightening with $K = 2$ NFEs, evaluated at time $t_0 = 0$ and time $t_2 = 0.4$. Blue arrows are velocity vectors given by the *pretrained* model, and purple arrows following the dashed lines are the ideal straight path. The straightening procedure in Section 4 aims to align the blue arrows towards the purple arrows. Arrows illustrate directions and are not drawn with proper scale.

sampling with $K^{\max} = 5$. This motivates the following alignment procedure:

$$\min_{\theta} \; \mathbb{E}_{X_0 \sim \pi_0} \Big[ \sum_{k=0}^{K-1} \Big\| v_{\theta}(X_{\tau_k}^{\mathcal{E}(K^{\max})}, \tau_k) - \frac{X_{\tau_{k+1}}^{\mathcal{E}(K^{\max})} - X_{\tau_k}^{\mathcal{E}(K^{\max})}}{\tau_{k+1} - \tau_k} \Big\|_2^2 \Big],$$

where $X_{\tau_k}^{\mathcal{E}(K^{\max})}$ is obtained by the Euler uniform sampling with $K^{\max}$ NFEs of the initial condition $X_0 \sim \pi_0$. Replacing the expectation with $n$ empirical paths obtained by $\mathcal{E}(K^{\max})$, we have the sample averaging optimization problem for straightening

$$\min_{\theta} \; \frac{1}{n} \sum_{i=1}^{n} \sum_{k=0}^{K-1} \| v_{\theta}(x_{\tau_k}^i, \tau_k) - \frac{x_{\tau_{k+1}}^i - x_{\tau_k}^i}{\tau_{k+1} - \tau_k} \|_2^2. \tag{9}$$

We straighten the velocity model using a stochastic gradient descent algorithm to solve the above problem. In the main paper, we train all parameters $\theta$ of the pretrained model, whereas in Appendix E, we employ Low-Rank Adaptations to reduce the number of trainable parameters while preserving the performance of the straightening process.

## 5 NUMERICAL EXPERIMENTS

**Settings.** We evaluate our methods on unconditioned image generation tasks. In particular, we use the CIFAR-10 (Krizhevsky et al., 2009) and three high-resolution (256x256) datasets CelebA-HQ (Karras et al., 2018), LSUN-Church, LSUN-Bedroom (Yu et al., 2015), and AFHQ-Cat. We take the checkpoints of pretrained velocity networks $v$ from the official implementation[1] of Rectified Flow (Liu et al., 2022b), which is based on the U-Net architecture of DDPM++ (Song et al., 2020b). If not mentioned otherwise, we evaluate the sample schemes with NFE=$\{4, 6, 8\}$. The quality of image samples is with Frechet inception distance (FID) score (Heusel et al., 2017).

**Baselines.** A comparison between Bellman optimal stepsizes and the conventional first/second order methods using uniform stepsizes is presented in Section 5.1. We also include an adaptive strategy, the Runge-Kutta method of order 5(4) from SciPy (Virtanen et al., 2020). In Section 5.2, our fine-tuning procedure, Bellman Optimal Stepsize Straightening (BOSS), is compared with two baselines including the Uniform-Reflow, and Distill-k-Reflow introduced in (Liu et al., 2022b).

### 5.1 IMPROVEMENTS OF FIRST ORDER AND SECOND ORDER SAMPLING SCHEME USING BELLMAN OPTIMAL STEPSIZE

First, we benchmark pretrained Euler samplers with uniform and Bellman optimal stepsizes, calculated following Section 3.3. The quantitative results are demonstrated in Table 1. The FID is much lower for samples generated by Euler's method with the Bellman step, which shows that with Bellman's optimal step size, the generated images are of much higher quality in general. Specifically, for sampling on the three larger dimension datasets (256x256), Bellman steps can help drastically reduce FID compared to the uniform step size. This trend is also reflected in Figure 5, which shows our qualitative results.

Table 1: FID ($\downarrow$) of Euler's sampling method with uniform stepsizes vs. Bellman optimal stepsizes on unconditional image generation task across different datasets.

| Dataset | 4 NFE | | 6 NFE | | 8 NFE | |
|---|---|---|---|---|---|---|
| | Euler | Bellman | Euler | Bellman | Euler | Bellman |
| CIFAR-10 | 51.95 | **47.57** | 25.69 | **23.35** | 16.82 | **15.74** |
| CelebA-HQ | 158.95 | **92.03** | 127.01 | **72.54** | 109.42 | **49.80** |
| LSUN-Church | 106.94 | **80.91** | 53.85 | **45.09** | 34.74 | **33.22** |
| AFHQ-Cat | 68.95 | **45.54** | 61.50 | **36.15** | 56.96 | **33.94** |
| LSUN-Bedroom | 84.35 | **61.60** | 39.19 | **35.35** | 32.15 | **25.80** |

We also empirically analyze the effect of Bellman optimal stepsizes on popular ODE solvers including Euler and Heun (second-order version). To avoid the confusion between schemes, we denote compared methods as follows:

---

[1] https://github.com/gnobitab/RectifiedFlow/

- Uniform Euler and Uniform Heun are the conventional Euler and Heun methods that use uniform sampling steps.
- Bellman Euler and Bellman Heun are two variants of the above methods, but using our proposed Bellman step sizes.
- RK45 is an adaptive strategy, the Runge-Kutta method of order 5(4) from Scipy.

The quantitative results are displayed in Figure 4. It is evident that employing Bellman optimal stepsizes can significantly improve FID scores (image quality) compared to using uniform stepsizes across all pretrained models on four distinct datasets. The Bellman Heun methods approach the performance of RK45 with substantially fewer NFEs. To elaborate, the Bellman Heun method achieves approximately a 1% gap compared to RK45 with just 20 sampling steps and fully recovers the performance of RK45 with 50 steps.

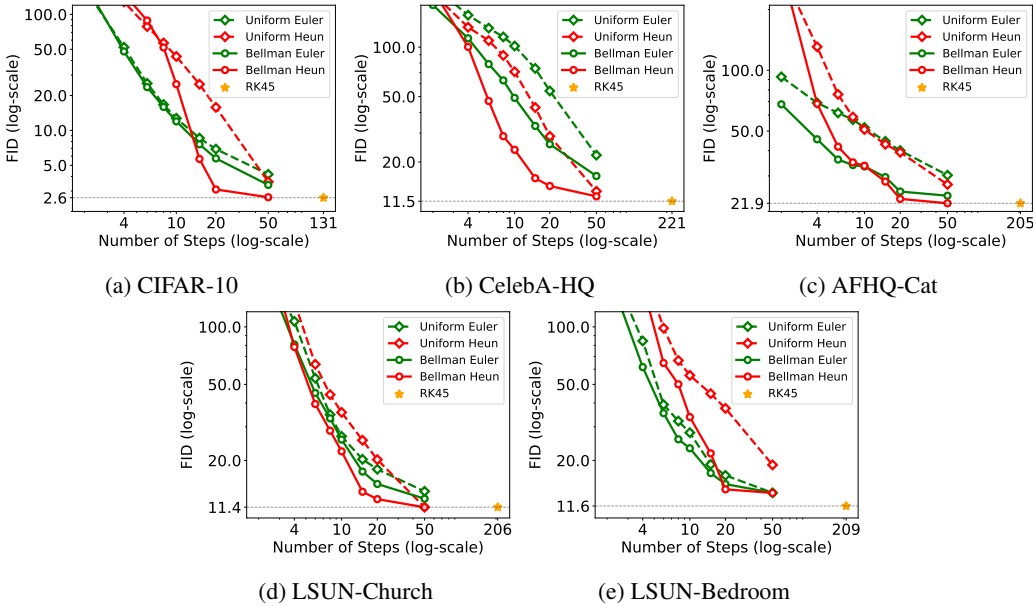

Figure 4: The FID score of sampling methods with different numbers of function evaluations (step sizes). Images generated by samplers using Bellman stepsizes clearly show lower FID than conventional ones that use uniform step sizes. Note that Uniform Heun and Bellman Heun are second-order sampling methods that use twice the NFEs.

## 5.2 EFFECTS OF REFLOW WITH BELLMAN STEPSIZE

After calculating the Bellman optimal step size, we follow the procedure described in Section 4 to straighten the probability path. The results, seen in Table 2, suggest that BOSS performs almost equally with the reflow procedure on CIFAR-10 but markedly better on the other four higher-dimension datasets. This is consistent with the visible improvements in sampled image quality observed in Figure 5.

Table 2: FID (↓) of different retraining methods on the unconditional image generation task across different datasets. Distill-K-Reflow is a distillation technique that relies on the reflow of the velocity network on a discrete grid of K uniform stepsizes between 0 and 1, as elaborated in Liu et al. (2022b).

| Dataset | Distill-6-Reflow | BOSS-6 | Uniform-Reflow |
|---|---|---|---|
| CIFAR-10 | 4.35 | 4.80 | **4.33** |
| CelebA-HQ | 34.56 | **18.67** | 43.57 |
| LSUN-Church | 34.52 | **17.43** | 40.45 |
| AFHQ-Cat | 46.24 | **26.10** | 51.24 |
| LSUN-Bedroom | 41.17 | **18.45** | 45.13 |

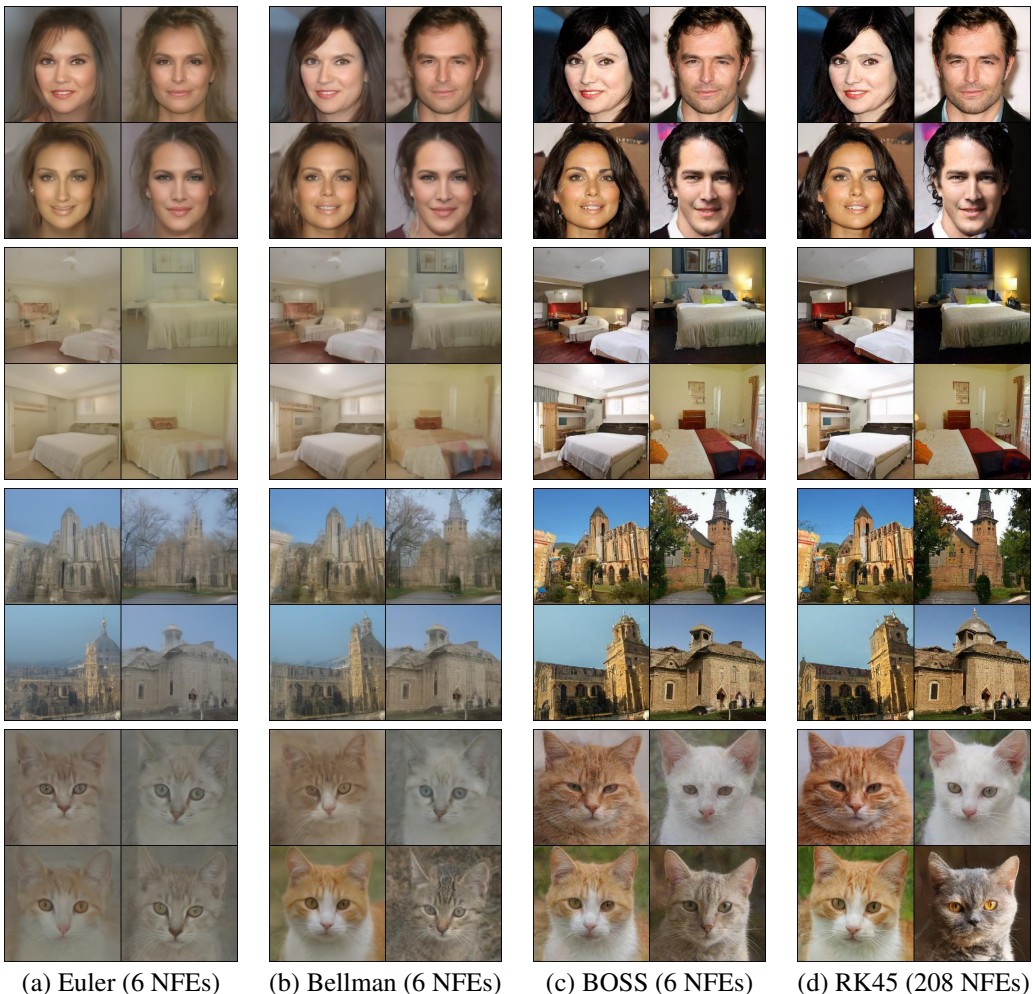

|        (a) Euler (6 NFEs)        |        (b) Bellman (6 NFEs)        |        (c) BOSS (6 NFEs)        |        (d) RK45 (208 NFEs)        |

Figure 5: Qualitative results on unconditional image generation task. From first to last row: CelebA-HQ/LSUN-Bedroom/LSUN-Church/AFHQ-Cat dataset. (a)-(b): Comparisons of Euler stepsizes between uniform (a) and the Bellman optimal stepsizes (b); (c)-(d): Comparisons of BOSS retraining and Runge-Kutta-45 sampling. Notice our proposed BOSS sampling has comparably similar visual quality to RK45 while requiring only 6 NFEs, compared to 208 NFEs of RK45.

# 6 CONCLUSIONS

This paper proposed BOSS, the Bellman Optimal stepsize Straightening method, to adapt pretrained flow-matching models under low computational resource constraints. Our method consists of two phases: first, find optimal sampling stepsizes for the pretrained model, then straighten out the velocity network on each interval of the sampling schedule. We demonstrate empirically that BOSS performs competitively in adapting pretrained models in the image generation task. Similar to training-based samplers for diffusion and flow matching models, a limitation of our method is the additional training cost to output the optimal sample step sizes. There are many potential extensions to our proposed framework to distill a guided velocity network, similar to Meng et al. (2023), or a computationally cheaper algorithm for calculating the Bellman sampling step sizes.

**Acknowledgments.** Viet Anh Nguyen gratefully acknowledges the generous support from the CUHK's Improvement on Competitiveness in Hiring New Faculties Funding Scheme and the CUHK's Direct Grant Project Number 4055191. The work of Binh Nguyen is supported by Singapore's Ministry of Education grant A-0004595-00-00.

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

## A    DETAILS OF EXPERIMENTS

We use the following checkpoints that are downloaded from the GitHub folder [2]:

- CIFAR-10: at iteration 800,000;

- CelebA-HQ: at iteration 1,000,000;

- LSUN-Church: at iteration 1,200,000;

- LSUN-Bedroom: at iteration 1,000,000;

- AFHQ-Cat: at iteration 1,000,000.

The pretrained models are finetuned in 12,000 iterations. One iteration is the passing and back-propagation process for a batch including 15 samples. Due to the similar cost of training between finetuning methods, we report the average GPU hours consumed on each pretrained model up to 12000 iterations, using NVIDIA RTX A5000.

- CIFAR-10: 3.56 training hours.

- CelebA-HQ: 10.35 training hours.

- LSUN-Church: 13.43 training hours.

- LSUN-Bedroom: 14.23 training hours.

- AFHQ-Cat: 9.30 training hours.

With this limited budget of resources, the proposed method, BOSS, achieves significantly better performance than other methods in terms of FID score. The value of $N$ in Equation (5) and $K^{\max}$ are fixed at 100, and 100 in all experiments if not mentioned. This setup $N = 100$ means we only use one batch sampling to calculate the truncation errors between timestamps in equation (5). It is worth noting that this setup highlights the low-cost and limited-resources requirement of our proposal.

**FID Calculation.**    All the FID metrics of related works are either cited from previous baselines or are calculated (when there are no such figures reported) based on the Clean-FID paper (Parmar et al., 2022), which unifies the FID calculation to make a fair comparison between papers. These four datasets were downloaded following the instructions from their original papers. We then create the stats file by the clean-fid project. The FID score is calculated based on 50,000 generated images and the stats file.

## B    DESCRIPTION OF THE DYNAMIC PROGRAMMING ALGORITHM

This section presents the pseudocode in Algorithm 1 for the practical implementation of the dynamic programming algorithm designed to determine the Bellman optimal stepsizes. The algorithm takes a cost matrix, denoted as $c$, as input, where $c_{jk}$ is computed using Equation (5) and the specified number of function evaluations (NFEs). The most resource-intensive aspect of this code is the nested loop responsible for calculating $\kappa(j, k)$, incurring a time complexity of $O((K^{\max})^2 \times K)$. The choice of $K^{\max}$ is crucial, aiming for the Euler sampling method with $K^{\max}$ stepsizes to precisely replicate the trajectory of the Ordinary Differential Equation (ODE). Typically, $K^{\max}$ falls within the range of 100 to 1,000, ensuring accuracy. Given this range for $K^{\max}$ and the variable $K$ ranging from 2 to 1,000, the algorithm executes within milliseconds in all scenarios.

---

[2]https://github.com/gnobitab/RectifiedFlow/

---

**Algorithm 1** Minimum Cost Path Computation

---

**Input:** Cost matrix $c(j,k) = c_{jk}(j,k = 0, \ldots, K^{\max})$ , the target NFEs $K$.
Set $\kappa(i,j) \leftarrow +\infty \quad \forall 0 \leq i \leq K^{\max}, 0 \leq j \leq K$
Set $\kappa(i,1) \leftarrow c(i,K^{\max}) \quad \forall 0 \leq i \leq K^{\max}$
**for** $k = 2$ **to** $K$ **do**
  **for** $j = 0$ **to** $K^{\max} - 1$ **do**
    **for** $i = j + 1$ **to** $K^{\max} - 1$ **do**
      $\kappa(j,k) \leftarrow \min\{\kappa(j,k), c(j,i) + \kappa(i,k-1)\}$
    **end for**
  **end for**
**end for**
Initialize $\psi \leftarrow [0], \omega \leftarrow 0$
**for** $k = K$ **to** $1$ **do**
  **for** $j = \omega + 1$ **to** $K^{\max}$ **do**
    **if** $\kappa(\omega,k) = c(\omega,j) + \kappa(j,k-1)$ **then**
      Append $j$ to $\psi$ and set $\omega \leftarrow j$
      **break**
    **end if**
  **end for**
**end for**
Append $K^{\max}$ to $\psi$.
**return** $\psi, \kappa(0,K)$

---

# C  EMPIRICAL ANALYSIS ABOUT BELLMAN OPTIMAL STEPSIZES

## C.1  A COMMON TREND IN BELLMAN OPTIMAL STEPSIZES FOR PRETRAINED MODELS ON DIFFERENT DATASETS

We plot the Bellman Optimal stepsizes in Figure 6. It shows a common trend of sampling with small initial steps and then larger stepsizes for intermediate iterations. At $K = 6$, we can observe that the last step is smaller than the penultimate step, hinting that the sampling process aims to take smaller final steps to refine the output. This refining trend is more evident for $K = 8$.

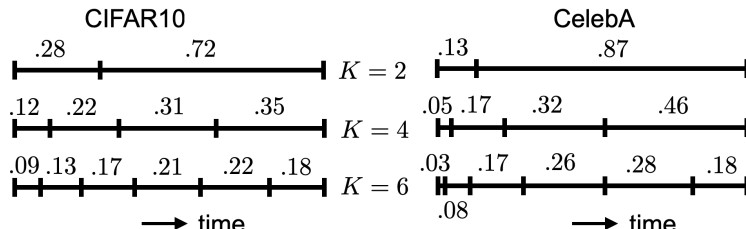

Figure 6: Optimal Bellman stepsizes for CIFAR-10 and CelebA-HQ at $K = 2, 4$ and $6$. One can identify a common pattern of smaller steps at the beginning and the end of the sampling procedure

## C.2  EMPIRICAL EVIDENCE FOR THE STEPSIZES TREND

This section aims to experimentally explain the trend of smaller stepsizes at the beginning and the end of the sampling procedure. Given a velocity network $v_\theta$, we empirically estimate the curvature using the following procedure:

1. Sample $N$ noise instances $x_0^i$ i.i.d. from a Gaussian distribution (we set $N \approx 1000$).

2. Forward each noise instance using $v_\theta$ to obtain $x_t^i$ for $t = 1, \ldots, K^{\max}$.

3. Compute the local curvature for each trajectory:

$$\text{Curv}_t^i = \|x_t^i - \frac{x_{t+1}^i + x_{t-1}^i}{2}\|_2^2 \quad \text{for } t = 2, \ldots, K^{\max} - 1.$$

4. Calculate the average curvature:

$$\widehat{\mathrm{Curv}}_t = \frac{1}{N} \sum_{i=1}^{N} \mathrm{Curv}_t^i \qquad \forall t = 2, \ldots, K^{\max} - 1.$$

Subsequently, we plot $\widehat{\mathrm{Curv}}_t$ and investigate whether the Bellman stepsizes coincide with the straightness of the curve, as demonstrated in Figure 7. The curvature of all pretrained rectified models is significant at timestamps close to zero (near the space of noises) and one (near the space of real images). This observation indicates a variant stepsize trajectory, notably at initial and final timestamps. It can be intuitively explained that additional steps are required to match the high-curvature region accurately. Consequently, the outcome of our proposal is sensible, as it correctly identifies the high curvature levels at the beginning and end of the sampling trajectories.

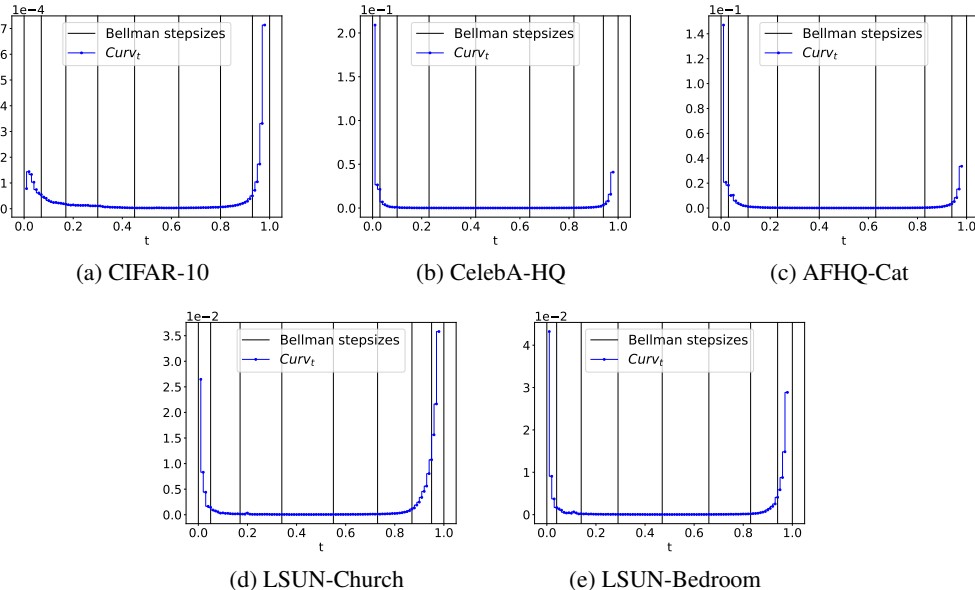

Figure 7: Curvature measurement (blue curve) and Bellman optimal timestamps (black horizontal line) for different datasets. We observe that the timestamps are denser at regions with higher estimated curvature.

## C.3 FINDING OPTIMAL STEPSIZES IS A LIGHT-WEIGHT PROCESS

In this section, we elaborate further on the efficiency of our framework. This efficiency arises because we only need to pass **a single batch** of noise through the forward process to obtain the values for each intermediate timestamp. Subsequently, we calculate the local truncation error for any two timestamps. These local truncation errors have in total $K^{\max} \times (K^{\max} - 1)/2$, and they can be efficiently stored without requiring large memory. The dynamic programming involved in this process is also time-efficient, as elaborated in the Appendix B. For added credibility, we provide a running time of the entire stepsizes calculation process with NFE = 10, detailing the running time of each component across all our datasets in Table 3. All running times are for the Nvidia A5000, an old-generation GPU launched in April 2021. The whole process takes around 115 seconds to complete for the 256x256 datasets.

Table 3: Running time (seconds) for different datasets

|  | CIFAR-10 | CelebA-HQ | LSUN-Church | LSUN-Bedroom | AFHQ-Cat |
|---|---|---|---|---|---|
| One batch forward | 8.5977 | 47.7143 | 47.4024 | 48.4324 | 45.5489 |
| Local truncation errors | 10.066 | 67.9143 | 67.5273 | 65.9875 | 66.7532 |
| Dynamic programming | 0.0134 | 0.0138 | 0.0138 | 0.0139 | 0.0138 |
| The whole process | 18.6771 | 115.6424 | 114.9435 | 114.4338 | 112.3159 |

Table 4: Bellman Optimal Stepsizes for $K = 2$ and $K = 4$. The total sum of stepsizes equals one.

|  | $K = 2$ | $K = 4$ |
|---|---|---|
| CIFAR-10 | $[0.28, 0.72]$ | $[0.12, 0.22, 0.31, 0.35]$ |
| CelebA-HQ | $[0.13, 0.87]$ | $[0.05, 0.17, 0.32, 0.46]$ |
| LSUN-Church | $[0.20, 0.80]$ | $[0.08, 0.21, 0.34, 0.37]$ |

Table 5: Bellman Optimal Stepsizes for $K = 6$ and $K = 8$. The total sum of stepsizes equals one.

|  | $K = 6$ | $K = 8$ |
|---|---|---|
| CIFAR-10 | $[0.09, 0.13, 0.17, 0.21, 0.22, 0.18]$ | $[0.07, 0.09, 0.12, 0.14, 0.17, 0.17, 0.14, 0.10]$ |
| CelebA-HQ | $[0.03, 0.08, 0.17, 0.26, 0.28, 0.18]$ | $[0.02, 0.05, 0.11, 0.17, 0.21, 0.20, 0.16, 0.08]$ |
| LSUN-Church | $[0.06, 0.15, 0.24, 0.26, 0.20, 0.09]$ | $[0.04, 0.09, 0.15, 0.17, 0.18, 0.17, 0.13, 0.07]$ |

The Bellman steps being far from uniform also means that the probability path of the pretrained models is far from straight, and performing the straightening operation would be beneficial.

We report in this section the Bellman optimal stepsizes obtained in Section 3.3. We want to focus on the case $K = 8$ to see the common trend of sampling step sizes. Initially, the sampling process takes small step sizes, possibly for structural determination of the images. The stepsizes become larger for the intermediate steps. The last two stepsizes show a decreasing trend: the sampling process takes small stepsizes at the end to refine and potentially make the final output less noisy.

## D  THE TRANSFER OF OPTIMAL STEPSIZES ACROSS DATASETS

To verify the generalization of optimized stepsizes, we transferred the optimized stepsizes from LSUN-Church to the pretrained models on CelebA-HQ and LSUN-Bedroom. The FID scores obtained with 4, 6, and 8 NFEs for CelebA-HQ resulting from this transfer are presented in Table 6.

Table 6: FID scores for CelebA-HQ with different methods and NFEs

| Method | 4 NFEs | 6 NFEs | 8 NFEs |
|---|---|---|---|
| Uniform Euler | 158.95 | 127.01 | 109.42 |
| Bellman Euler | 92.03 | 72.54 | 49.80 |
| Bellman-transfer | 132.04 | 100.68 | 72.88 |

Uniform Euler uses uniform step sizes, Bellman Euler uses optimal stepsizes for CelebA-HQ, while Bellman-transfer uses the stepsizes taken from LSUN-Church. Bellman Euler is still the optimal method. However, what is important here is that Bellman-transfer is better than Uniform Euler. This hints that there is a certain degree of transferability of the step sizes.[3] In the empirical realm, we can attribute this transferability to a comparable curvature pattern exhibited by pretrained rectified models, as discussed in Section C.2.

---

[3]This is an empirical claim; we do not impose any theoretical claim.

Table 7 is for the LSUN-Bedroom dataset. We observe the same trend here, empirically confirming that the stepsizes have a certain degree of transferability. Nevertheless, optimizing the stepsizes using Bellman Euler would still obtain the best performance.

Table 7: FID scores for LSUN-Bedroom with different methods and NFEs

| Method | 4 NFE | 6 NFE | 8 NFE |
|---|---|---|---|
| Uniform Euler | 84.35 | 39.19 | 32.15 |
| Bellman Euler | 61.60 | 35.35 | 25.80 |
| Bellman-transfer | 70.23 | 38.01 | 29.14 |

# E LOW-RANK ADAPTATION FOR STRAIGHTENING

In this section, we expertiment with adding a low-rank adaptation to the linear and convolutional layers of the velocity network $v_\theta$ . For instance, the $t^{\text{th}}$ linear layer represented by an $m \times n$ matrix $W_t$ is adapted to

$$\widehat{W_t} = W_t + A_t B_t^\top,$$

where $A_t$ is an $m \times r$ matrix, and $B_t$ is an $n \times r$ matrix. The value $r \ll \min\{m, n\}$ represents the rank of the adaptation. This adaptation is similarly applied to convolutional layers, with a slight adjustment: a convolutional layer is first reshaped into a two-dimensional matrix before incorporating the adaptation term. We keep all original parameters of the models fixed and only update the $A_t$ and $B_t$ matrices during the straightening process. We have four versions of straightening named LoRA-$r$, where $r$ is chosen from the set $\{1, 4, 16, 64\}$. Their FID scores on the CelebA-HQ dataset over training iterations are plotted in Figure 8, while the number of trainable parameters for each method is presented in Table 8. The FULL-RANK variant described in Section 4 is the method that finetuning *all* parameters of the original model. It is noticeable that versions LoRA-4, LoRA-16, and LoRA-64 can almost match the FID score of the FULL-RANK version, which is 33.86. Specifically, at 175,000 training iterations, LoRA-4, LoRA-16, and LoRA-64 achieve FID scores of 36.70, 34.52, and 34.20 respectively. Furthermore, LoRA-4 only finetunes 2% of the parameters of the original model but still achieves competitive results compared to the FULL-RANK version.

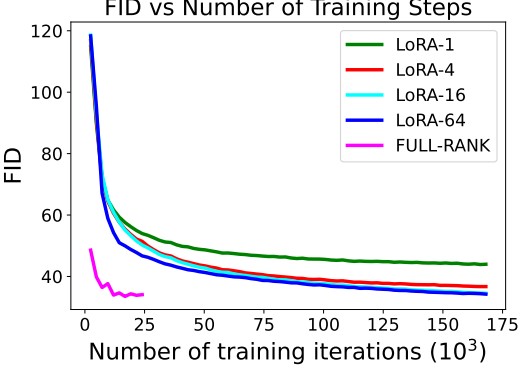

Figure 8: The FID score of straightening procedures along the training iterations on the CelebA-HQ dataset.

Table 8: The number of trainable parameters of straightening procedures compared to the full-rank straightening on the CelebA-HQ dataset

| Method | Number of Trainable parameters | Percentage on the FULL-RANK version (%) |
|---|---|---|
| LoRA-1 | 330,562 | 0.5 |
| LoRA-4 | 1,322,248 | 2.02 |
| LoRA-16 | 5,288,992 | 8.07 |
| LoRA-64 | 21,155,968 | 32.26 |
| FULL-RANK | 65,574,549 | 100 |

## F ADDITIONAL QUALITATIVE RESULTS

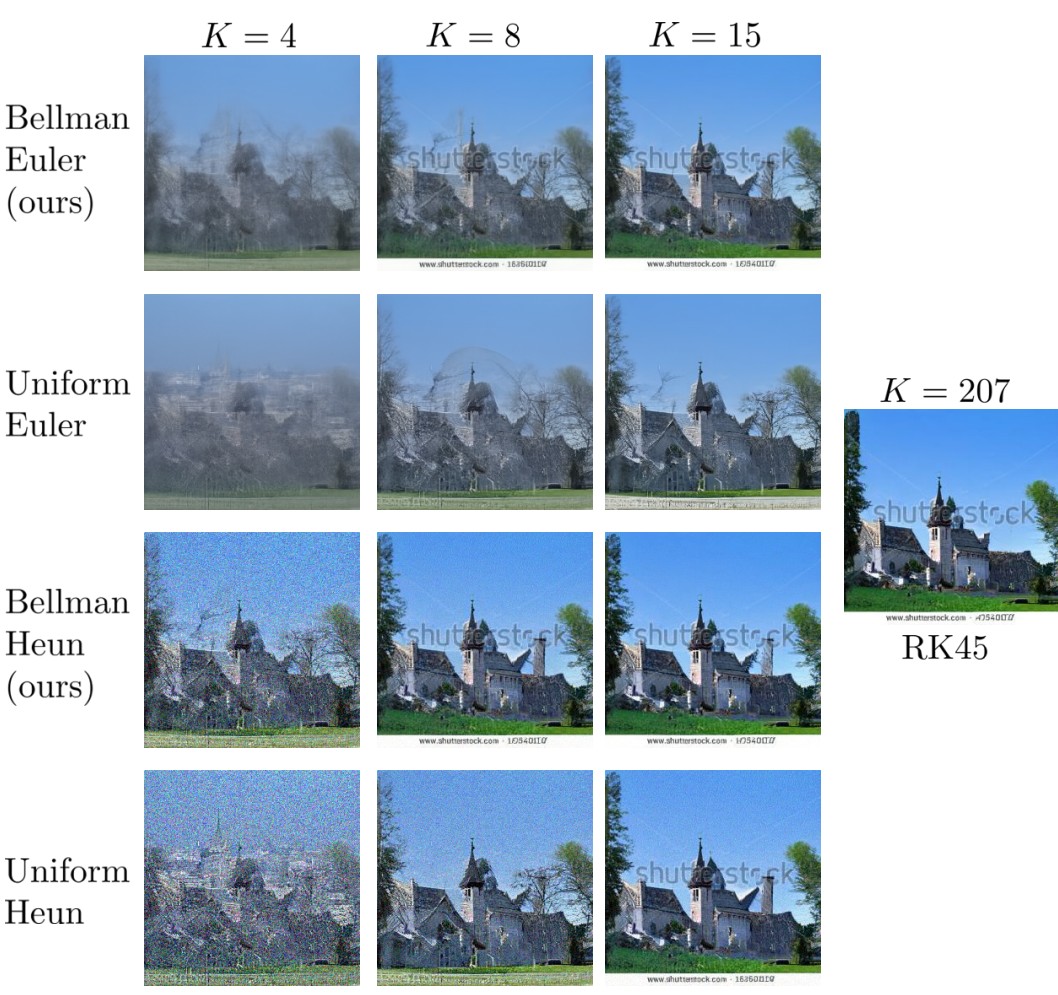

Figure 9: Comparison between images generated from an identical noise with different sampling methods and the number of stepsizes on the LSUN-Church dataset.

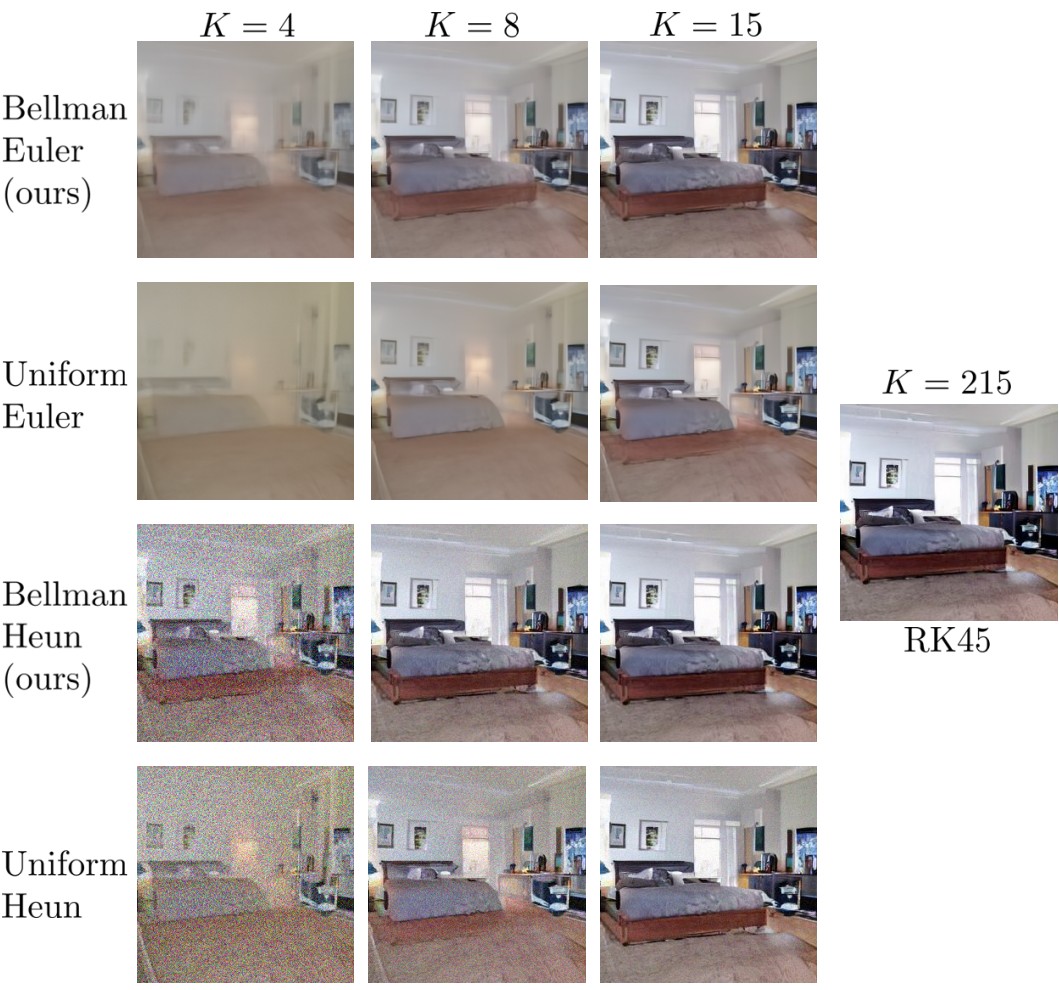

Figure 10: Comparison between images generated from an identical noise with different sampling methods and the number of stepsizes on the LSUN-Bedroom dataset.

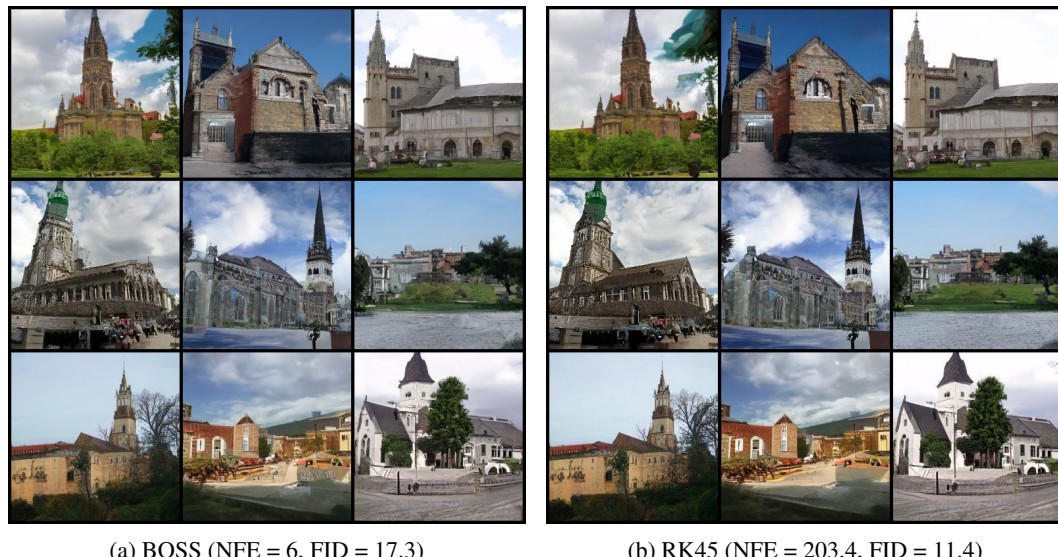

(a) BOSS (NFE = 6, FID = 17.3)          (b) RK45 (NFE = 203.4, FID = 11.4)

Figure 11: Comparative qualitative outcomes of BOSS with NFE = 6. The image on the right showcases the generated images referenced by RK45.

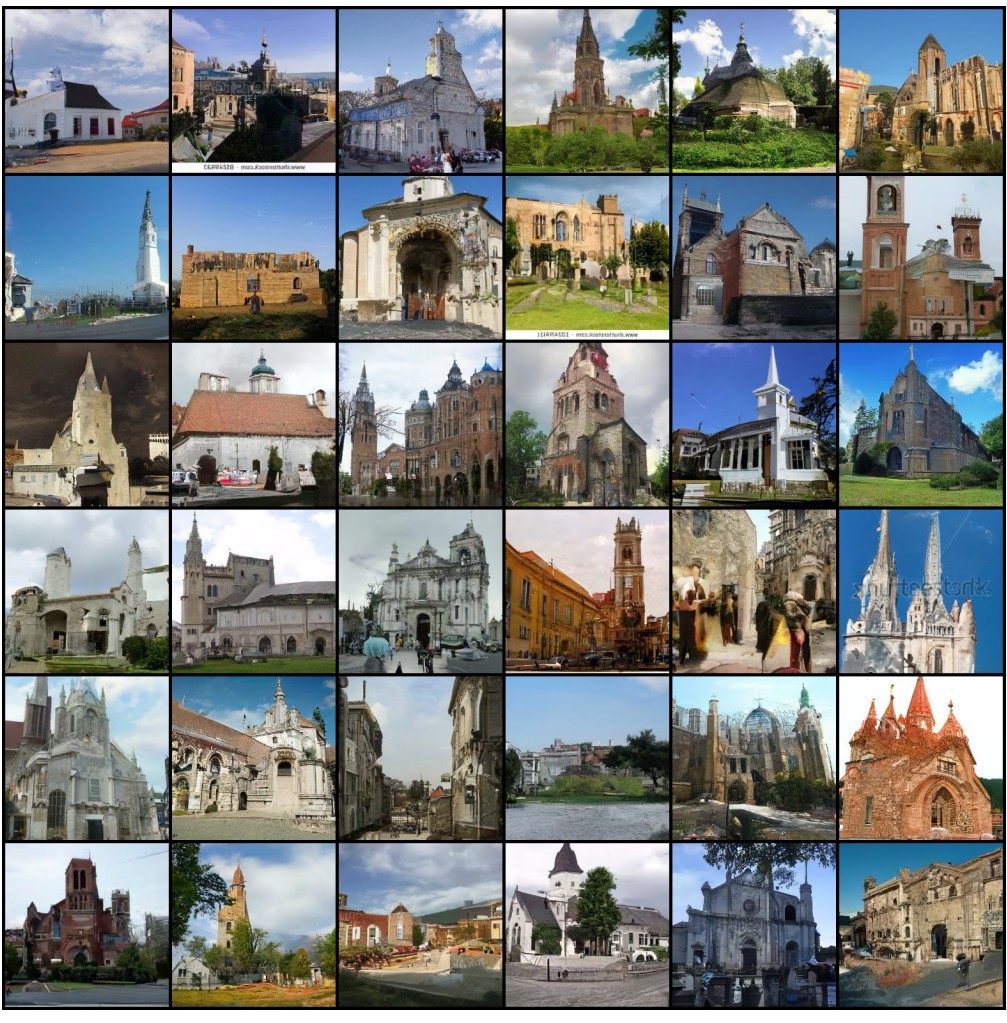

Figure 12: Uncurated images generated from the model finetuned by BOSS (NFE = 10, FID=13.89)

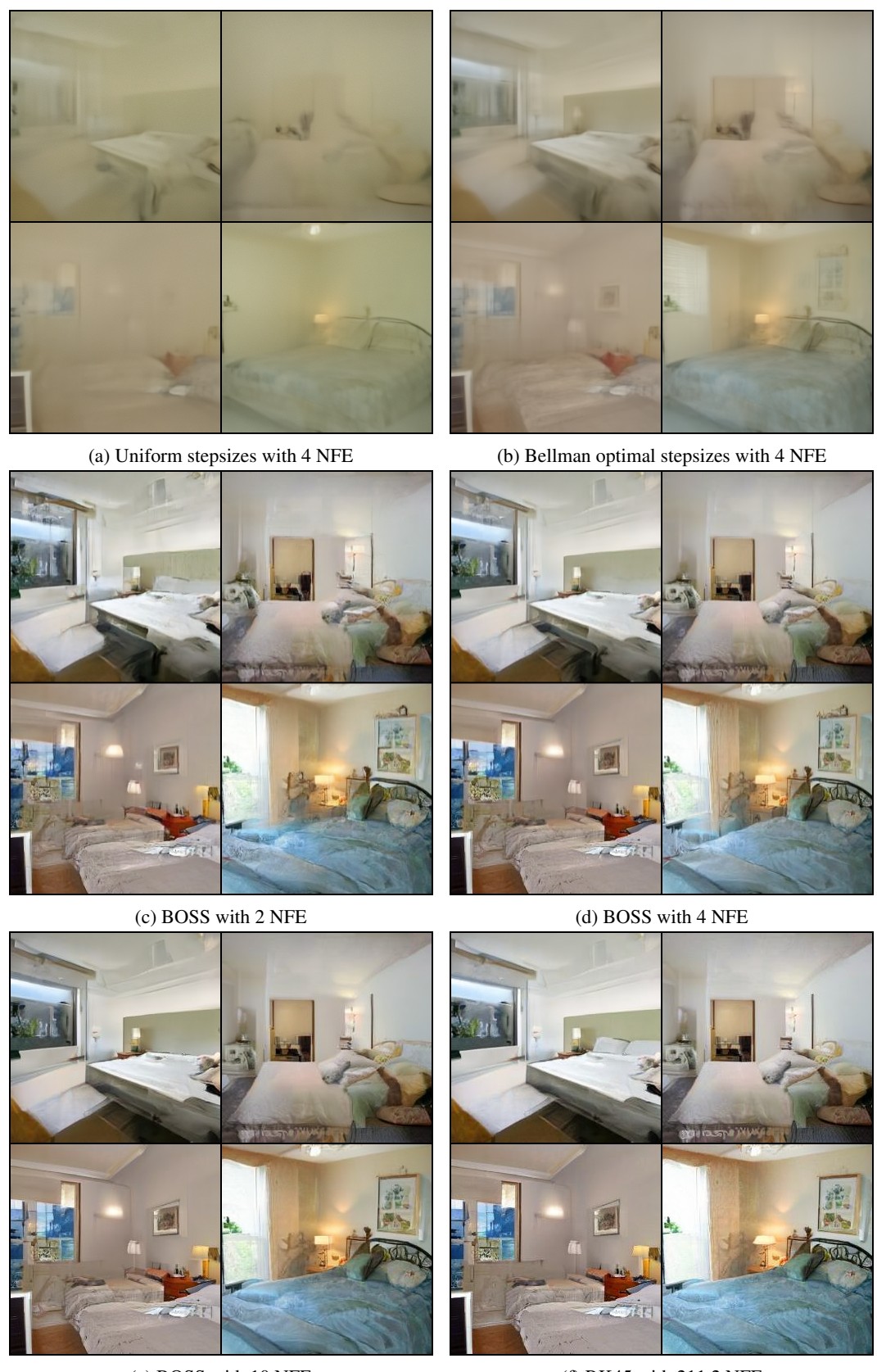

(a) Uniform stepsizes with 4 NFE        (b) Bellman optimal stepsizes with 4 NFE

(c) BOSS with 2 NFE        (d) BOSS with 4 NFE

(e) BOSS with 10 NFE        (f) RK45 with 211.2 NFE

Figure 13: Comparative qualitative outcomes of BOSS with different NFEs on the LSUN-Bedroom dataset.

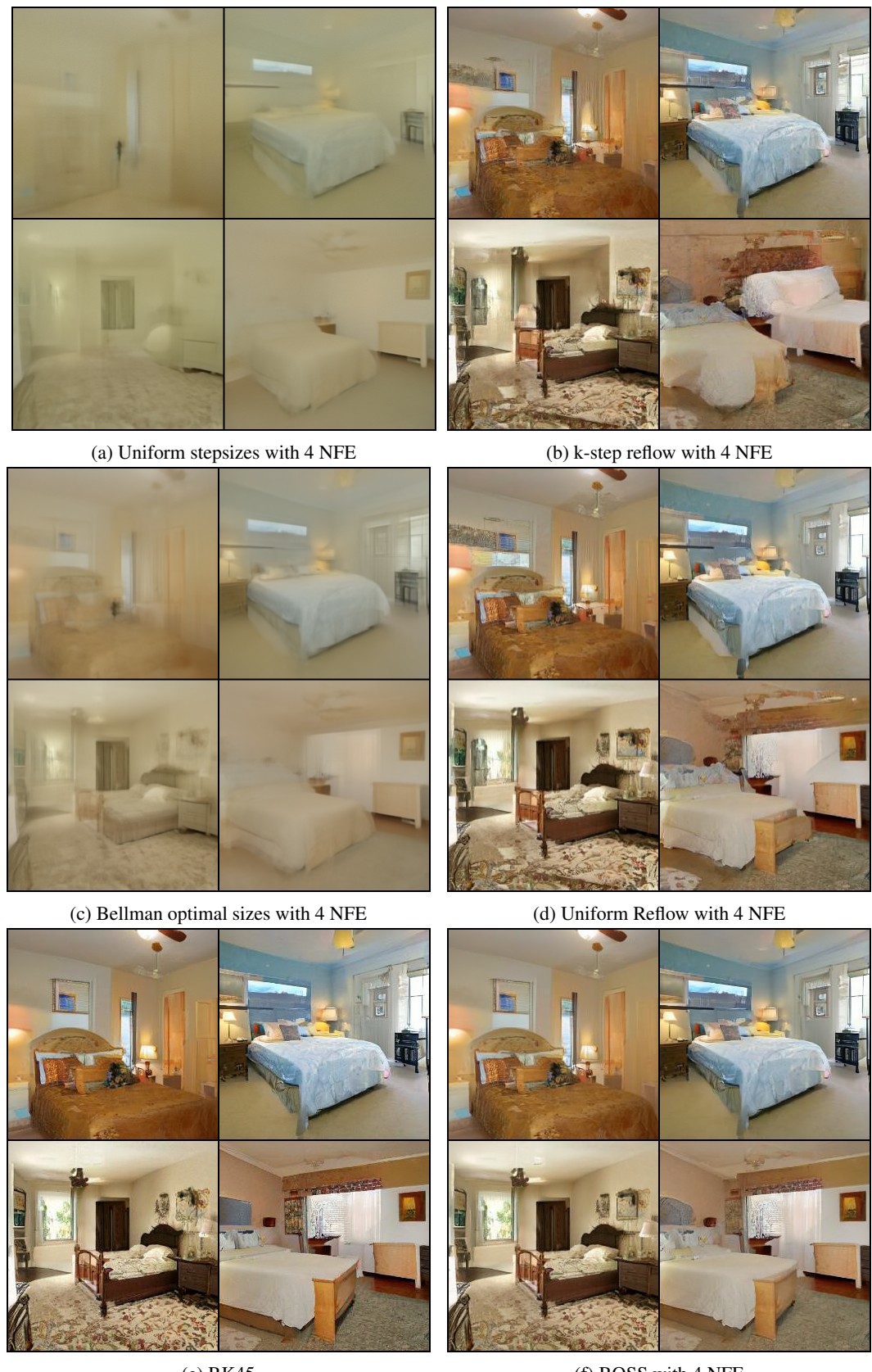

Figure 14: Samples from LSUN-Bedroom. All corresponding samples use the same initial noise.

