# OpenReview forum: "Bellman Optimal Stepsize Straightening of Flow-Matching Models"
_ICLR.cc/2024/Conference — ICLR 2024 poster_

### Official Review · Reviewer_GU1Q · 2023-10-23

**Soundness:** 2 fair
**Presentation:** 3 good
**Contribution:** 3 good
**Rating:** 6
**Confidence:** 4

**Summary:**

This paper proposed a new sampling strategy for flow-matching (flow-reflected) generative models. Formally, the authors analyze the sampling error estimation and propose to formulate the sampling process as a dynamic programming problem. Then the task is addressed by the bellman optimal schedule. Moreover, the authors proposed to re-align the velocity network along the accumulated sampling errors, which enjoys good performance.

**Strengths:**

1. The authors clearly present the issue of sampling error in flow-matching generation.
2. It is interesting and intuitive to formulate the sampling schedule as a dynamic programming problem. The solution is convincing.
3. This paper is well-written with clear motivations.

**Weaknesses:**

1. Although the main idea of this paper is interesting, this paper gives me the initial impression of being incomplete, with this incompleteness of the presentation of both the methodology and experimental sections.
2. For method Sec.3.3, the authors did not provide enough details about how to address the dynamic programming (DP) problem in practice. From the experiments, different datasets should share different sampling schedules. Moreover, how many samples are used to calculate the optimal scheduler? What will the optimal scheduler be like, if using different initial noises? The analysis of the generalization of the optimal scheduler selected by DP is very important.
3. Missing necessary discussions about diffusion sampling schedules in related works (such as [1]). Moreover, the related works are too simple, and only include some matching-flow methods.
3. Insufficient qualitative comparisons: the authors only provide qualitative comparisons of human faces.
4. No details about the velocity network re-alignment training are provided.

[1] Karras T, Aittala M, Aila T, et al. Elucidating the design space of diffusion-based generative models[J]. Advances in Neural Information Processing Systems, 2022, 35: 26565-26577.

**Questions:**

Although this paper enjoys interesting idea, it suffers from incomplete related works, experimental results, analysis, and methodology/implementation details, which largely limit the quality of this paper. I think this paper needs a major revision before the publication.

---

> ### Author Response · Authors · 2023-11-20
> **Reponse to Reviewer GU1Q**
>
> > Although the main idea of this paper is interesting, this paper gives me the initial impression of being incomplete, with this incompleteness of the presentation of both the methodology and experimental sections.
>
> We appreciate the reviewer’s comment and have added various elaborations in the main text. Please refer to the improved version of our submission with added elaborations, new results with additional datasets.
>
> > For method Sec.3.3, the authors did not provide enough details about how to address the dynamic programming (DP) problem in practice.
>
> We have added a section in the Appendix B with a pseudocode that provides details on our Dynamic Programming algorithm. The time complexity of this algorithm is $O((K^{max})^2 \times K)$. The choice of $K^{max}$ is crucial, aiming for the Euler sampling method with $K^{max}$ stepsizes to precisely replicate the trajectory of the Ordinary Differential Equation (ODE). Typically, $K^{max}$ falls within the range of 100 to 1000, ensuring accuracy. Given this range for $K^{max}$ and the variable $K$ ranging from 2 to 1000, the algorithm executes within seconds in all scenarios. Moreover, our solved problem is fast sampling in flow-matching models which focuses on relatively small values of K (<20), so the intensity of the dynamic programming algorithm is not a worrying problem even in the case that $K^{max} > 1000$.
>
> > From the experiments, different datasets should share different sampling schedules. Moreover, how many samples are used to calculate the optimal scheduler? What will the optimal scheduler be like, if using different initial noises? The analysis of the generalization of the optimal scheduler selected by DP is very important.
>
> We have added a section in the Appendix A on experimental details. In short, we used 100 samples for calculating the Bellman steps for all dataset, as we did not observe any improvements in FID when using more than that number of samples for calculating optimal step-size scheduler. Moreover, we aim to use as low computational resources as possible to keep it inline with the goal of our work.
> We agree with the reviewer that an analysis of the generalization of the optimal scheduler/sampler with DP steps is very important. However, we think it requires setting up plenty of additional definitions and assumptions related to learning theory (e.g. neural network class of the velocity networks, assumption about data and noise distributions, smoothness of the velocity functions, etc.) and therefore we consider this beyond the scope of the current work. However, we find this a very interesting direction to explore in the future work, and we thank the reviewer for helpful comment.
>
>
> > Missing necessary discussions about diffusion sampling schedules in related works. Moreover, the related works are too simple, and only include some matching-flow methods.
>
> We thank the reviewer for pointing this out, and we have updated the related work sessions accordingly. In short, the focus of our work is on improving the ODE sampler of flow-matching models. While flow matching is a special case of diffusion probability ODE flow, flow matching having different formulations for its probability path (linear interpolation $X_t = (1-t)X_0 + t X_1$ makes it possible for redressing/reflowing to work. The principle of measuring the straightness of the trajectory curve is not applicable to general affine formulation from diffusion probability ODE. Nevertheless, we think that extending the philosophy of calculating optimal step sizes/noise schedulers with dynamic programming is an interesting direction, and we will leave it for future works.
>
> > Insufficient qualitative comparisons: the authors only provide qualitative comparisons of human faces.
>
> We have added more detailed qualitative comparisons in the main text as suggested by the reviewer. Please refer to [Figure 5].
>
> > No details about the velocity network re-alignment training are provided.
>
> We are unsure if we understood correctly that the reviewer asked for details and training objective of the straightening velocity network. If this is the case, then the reviewer can refer to Section 4. On the other hand, if instead the reviewer wanted to know more about the details of velocity network retraining (GPU hours, training iterations, etc.), please kindly refer to our Appendix A (Experimental details). Below is a copy of the paragraph.
>
> The pre-trained models are finetuned by three methods including Uniform-Reflow, Distill-k Reflow, and BOSS  in 12000 iterations. An iteration is the passing and backpropagation process for a batch including 15 samples. Due to the similar cost of training between finetuning methods, we report the average GPU hours consumed on each pre-trained model up to 12000 iterations, using NVIDIA RTX A5000.
>
> * CIFAR10: 3.56 training hours.
> * CelebA-HQ: 10.35 training hours.
> * LSUN-CHURCH: 13.43 training hours.
> * LSUN-BEDROOM: 14.23 training hours.
>  * AFHQ-CAT: 9.30 training hours.

---

> > ### Comment · Reviewer_GU1Q · 2023-11-22
> > **Thanks for the response from the authors**
> >
> > The authors have extensively revised their paper, incorporating numerous details that have significantly enhanced its overall quality.
> >
> > However, a concern arises from the fact that all step-size selections are tailored to specific datasets, posing a challenge to achieving generalization. The absence of a universal approach may limit the broader applicability of the proposed methodology.
> >
> > To address this potential limitation, conducting additional empirical experiments would be beneficial on this issue. For instance, exploring the consequences of transferring the scheduler optimized for one dataset to another could provide valuable insights. Understanding whether such a transfer adversely affects performance would contribute to a more comprehensive evaluation of the proposed approach.

---

> ### Author Response · Authors · 2023-11-22
>
> Thanks for your constructive comments. We appreciate your suggestion of analyzing the transferability between stepsizes optimized for a dataset to other datasets. However, we argue that stepsize transferability is not a must for the following reasons:
>
> 1. In the current state of generative AI, for two different datasets, we have two different velocity networks. Each network is fine-tuned to generate samples only for the training dataset. If we have two different networks, there is no clear reason why we should use the same stepsize. Thus, stepsize transferability is not needed.
>
> 2. Our objective is not to find a common step size for every dataset. Our goal is to *customize* the stepsizes to the dataset so that we can reduce the number of steps while retaining the quality of the generated samples. We have demonstrated in our experiments that this can be done efficiently under low resources, even for moderately high resolution datasets such as CelebA-HQ 256, LSUN Church/Bedroom and AFHQ-Cat.
>
> The important message is that the *step size is not generalizable, but our framework is generalizable*. When applied to various datasets and pre-trained models, our framework consistently exhibits improvements over the baselines (uniform stepsizes, etc.). Thus, *our framework is universal*, at least for all the experiments in our paper.
>
> Nevertheless, we still follow the reviewer’s suggestion to check the transferability of dynamic programming step sizes, we agree that it can provide us further insights into our framework. Due to time constraints, we transfer the optimized step sizes from LSUN-CHURCH to the pre-trained models on CeleA-HQ and LSUN-BEDROOM.
> The FID scores, obtained with 4, 6, and 8 NFEs, for CelebA-HQ resulting from this transfer are presented in the tables below.
>
> | Method         | 4 NFE   | 6 NFE   | 8 NFE   |
> |-------------------|---------|---------|---------|
> | Uniform Euler             | 158.95  | 127.01  | 109.42  |
> | Bellman Euler           | 92.03   | 72.54   | 49.80   |
> | Bellman-transfer  | 132.04  | 100.68  | 72.88   |
>
> Uniform Euler uses uniform stepsizes, Bellman Euler uses optimal stepsizes for the CelebA-HQ, while Bellman-transfer uses the stepsizes taken from the LSUN-CHURCH. It is clear that Bellman Euler is still the optimal method. However, what is important here is that Bellman-transfer is *better* than Uniform Euler. This hints that there is a certain degree of transferability of the stepsizes (this is an empirical claim, we do not impose any theoretical claim at this point).
>
> The below table is for the LSUN-BEDROOM:
> | Method       | 4 NFE   | 6 NFE   | 8 NFE   |
> |--------------------|---------|---------|---------|
> | Uniform Euler              | 84.35   | 39.19   | 32.15   |
> | Bellman Euler           | 61.60   | 35.35   | 25.80   |
> | Bellman-transfer   | 70.23   | 38.01   | 29.14   |
>
> We observe the same trend here, confirming empirically that there is a certain degree of transferability in the stepsize. Nevertheless, if we optimize the stepsize and use Bellman Euler, we would still obtain the best performance.
>
> These empirical results reinforce the importance of *customizing* stepsizes to each dataset to obtain superior performance.
> Once again, finding the optimal stepsizes is efficient using our proposed framework.
>
> Let us elaborate further on the efficiency of our framework:. This efficiency arises because we only need to pass *a single batch* of noise through the forward process to obtain the values for each intermediate timestamp. Subsequently, we calculate the local truncation error for any two timestamps. These local truncation errors has in total $K^{\max}*(K^{\max}-1)/2$ and they can be efficiently stored without requiring large memory. The dynamic programming involved in this process is also time-efficient, as elaborated in the Appendix. For added credibility, we provide a running time of the entire stepsizes calculation process with NFE = 10, detailing the running time of each component across all our datasets in the below table. All running time are for Nvidia A5000 (an old generation GPU, launched in April 2021), CPU: Intel Xeon 2.4-3.7 GHz, 20 core, 40 thread. The whole process just takes around 115 seconds to complete for the 256x256 datasets.
>
> | Running time (s)        | CIFAR10  | CelebA-HQ | LSUN-CHURCH | LSUN-BEDROOM | AFHQ-CAT  |
> |--------------------------|----------|-----------|-------------|--------------|-----------|
> | One batch forward        | 8.5977   | 47.7143   | 47.4024     | 48.4324      | 45.5489   |
> | Local truncation errors  | 10.066   | 67.9143   | 67.5273     | 65.9875      | 66.7532   |
> | Dynamic programming      | 0.0134   | 0.0138    | 0.0138      | 0.0139       | 0.0138    |
> | The whole process        | 18.6771  | 115.6424  | 114.9435    | 114.4338     | 112.3159  |

---

> > ### Author Response · Authors · 2023-11-23
> > **A reminder**
> >
> > Dear reviewer,
> >
> > It is now only several hours until the author-reviewer discussion period ends. It would be really great if you can acknowledge that you have read the rebuttal and our revised version of the work. If you have any additional questions or concerns, we will try our best to address them quickly. Otherwise, we would appreciate if you can adjust your evaluation accordingly. Thank you for your valuable feedback, and we sincerely anticipate your response.
> >
> > Best regards, The authors.

---

> > ### Comment · Reviewer_GU1Q · 2023-11-23
> > **Thanks for the further discussion**
> >
> > I understand the concern of authors that different models are trained on different datasets, and they should share different step sizes. However, it is crucial that the overarching objective of this method extends to large diffusion models, such as the recently proposed InstaFlow. I acknowledge the inherent difficulty in evaluating the performance of such expansive matching flow models during the rebuttal phase. Consequently, I suggest that the authors delve deeper into discussions and analyses pertaining to the generalization of their method.
> >
> > On the whole, the authors have effectively addressed all my concerns. The only notable observation is that this paper has undergone a substantial revision compared to its initial submission, encompassing essential experiments and discussions. If the AC deems this extensive alteration acceptable, I would be inclined to revise my score to borderline acceptance.

---

### Official Review · Reviewer_pSht · 2023-10-31

**Soundness:** 3 good
**Presentation:** 3 good
**Contribution:** 2 fair
**Rating:** 6
**Confidence:** 5

**Summary:**

Flow matching is a powerful framework for generating high-quality samples, especially in image synthesis. However, the computational demands of these models pose challenges in low-resource scenarios. This paper introduces the Bellman Optimal Step-size Straightening (BOSS) technique for efficient image sampling within a computational budget. BOSS optimizes step sizes and refines the velocity network to improve generation paths. Experimental evaluations demonstrate BOSS's effectiveness in resource utilization and image quality. It provides a sustainable solution that reduces costs and environmental footprints.

**Strengths:**

Not only in flow matching, but also in diffusion models, all endeavors are focused on predicting the efficiency of noise, score, and vector field. While most of them set the step size based on a heuristic principle (e.g., DDIM, DDPM), this paper explores generative models of ODE from a different perspective - the step size itself, rather than the "direction" of a specific step.

**Weaknesses:**

- The baseline is too weak, as it only compares quantitatively with the fixed-step size ODE solver euler, while it doesn't compare with adaptive step size ODE solvers such as dopri5 or rk45. This comparison should be included in Table 1.
- The optimization needs to be conducted on a case-by-case basis. Additionally, the use of the dynamic programming algorithm using Gorubi may result in slow performance, rendering the method impractical. Optimization time is also not discussed in this paper.

**Questions:**

as above

---

> ### Author Response · Authors · 2023-11-20
> **Reponse to Reviewer pSht**
>
> > The baseline is too weak, as it only compares quantitatively with the fixed-step size ODE solver euler, while it doesn't compare with adaptive step size ODE solvers such as dopri5 or rk45. This comparison should be included in Table 1.
>
> We thank the reviewer for the suggestion. We have added more baseline methods and additional datasets to showcase the effectiveness. The quantitative results can be found in [Table 1 & 2], Figure 4 of our revision of the paper, and the qualitative results can be found in Figure 5 of our paper.
>
> > The optimization needs to be conducted on a case-by-case basis. Additionally, the use of the dynamic programming algorithm using Gorubi may result in slow performance, rendering the method impractical.
>
> We agree that our method requires solving additional dynamic programming problem to find sampling step sizes, compared to the many other training-free diffusion/flow matching samplers that have closed-form formulation. However, we want to point out that in most cases, there are no guarantee  that these closed-form formulas are optimal, as they are mostly on heuristic rules.
>
> > Optimization time is also not discussed in this paper.
>
> Indeed this is missing from the previous version of our work. We have added a paragraph in the Appendix A & B regarding CPU+GPU hours of calculating Bellman steps and GPU hours of retraining/redressing the pretrained velocity networks.

---

> ### Author Response · Authors · 2023-11-22
>
> > Optimization time is also not discussed in this paper.
>
> Let us elaborate further on the efficiency of our framework:. This efficiency arises because we only need to pass *a single batch* of noise through the forward process to obtain the values for each intermediate timestamp. Subsequently, we calculate the local truncation error for any two timestamps. These local truncation errors has in total $K^{\max}*(K^{\max}-1)/2$ and they can be efficiently stored without requiring large memory. The dynamic programming involved in this process is also time-efficient, as elaborated in the Appendix. For added credibility, we provide a running time of the entire stepsizes calculation process with NFE = 10, detailing the running time of each component across all our datasets in the below table. All running time are for Nvidia A5000 (an old generation GPU, launched in April 2021), CPU: Intel Xeon 2.4-3.7 GHz, 20 core, 40 thread. The whole process just takes around 115 seconds to complete for the 256x256 datasets.
>
> | Running time (s)        | CIFAR10  | CelebA-HQ | LSUN-CHURCH | LSUN-BEDROOM | AFHQ-CAT  |
> |--------------------------|----------|-----------|-------------|--------------|-----------|
> | One batch forward        | 8.5977   | 47.7143   | 47.4024     | 48.4324      | 45.5489   |
> | Local truncation errors  | 10.066   | 67.9143   | 67.5273     | 65.9875      | 66.7532   |
> | Dynamic programming      | 0.0134   | 0.0138    | 0.0138      | 0.0139       | 0.0138    |
> | The whole process        | 18.6771  | 115.6424  | 114.9435    | 114.4338     | 112.3159  |

---

> > ### Author Response · Authors · 2023-11-23
> > **A reminder**
> >
> > Dear reviewer,
> >
> > It is now only several hours until the author-reviewer discussion period ends. It would be really great if you can acknowledge that you have read the rebuttal and our revised version of the work. If you have any additional questions or concerns, we will try our best to address them quickly. Otherwise, we would appreciate if you can adjust your evaluation accordingly. Thank you for your valuable feedback, and we sincerely anticipate your response.
> >
> > Best regards, The authors.

---

### Official Review · Reviewer_TqqW · 2023-11-06

**Soundness:** 3 good
**Presentation:** 3 good
**Contribution:** 3 good
**Rating:** 6
**Confidence:** 3

**Summary:**

The author introduces the Bellman Optimal Step-size Straightening (BOSS) technique, a method for distilling flow-matching generative models that enhances the efficiency of image sampling within computational budget constraints. BOSS utilizes dynamic programming to optimize step sizes in a pretrained network and refines the velocity network to straighten generation paths. In this paper, the proposed method has been extensively evaluated on image generation tasks, showing that it significantly improves resource efficiency and maintains high image quality. This approach serves to reconcile the intensive computational demands of flow-matching models with low-resource availability, contributing to the sustainable development of artificial intelligence by reducing computational expenses and environmental impacts.

**Strengths:**

The BOSS (Bellman Optimal Step-size Straightening) method presents an innovative two-phase approach for adapting pretrained flow-matching models. This paper illustrates that BOSS can straighten the velocity network with approximately 10,000 retraining iterations, which marks a significant improvement in efficiency compared to standard practices. Consistently, BOSS achieves lower FID scores in the task of unconditional image generation across a variety of datasets, suggesting superior image quality relative to competing methods. Moreover, the paper introduces a distinctive methodology for calculating optimal sampling step sizes through dynamic programming, thereby increasing the sampling process's efficiency.

**Weaknesses:**

Although the paper demonstrates significant improvements in image quality and efficiency, its testing is concentrated on specific datasets. A more comprehensive comparative analysis with current state-of-the-art methods would elucidate the advancements BOSS provides, especially in efficiency and quality. Additionally, while the paper addresses low-resource scenarios, it lacks a clear comparison of resource requirements such as memory usage, power consumption, or processing time, essential for evaluating BOSS's practicality for users with limited computational resources. Furthermore, despite the method's enhancements over existing approaches, the computational intensity of the dynamic programming algorithm and the network retraining requirement could limit its utility in resource-constrained settings. The scalability of BOSS, with respect to increasing dataset sizes or complexity, is also not addressed, and an evaluation of this aspect would greatly benefit the paper's comprehensiveness.

**Questions:**

Could the authors extend their testing to include a broader range of datasets and perform a comprehensive comparative analysis with current state-of-the-art methods to better highlight the efficiency and quality improvements of the BOSS method?
Can the paper provide a detailed comparison of resource requirements, such as memory usage, power consumption, or processing time, to evaluate the practicality of BOSS for users with limited computational resources?
How does the computational intensity of the dynamic programming algorithm and the network retraining requirement impact the method's applicability in resource-constrained environments?
 Additionally, has the scalability of BOSS been assessed in relation to increasing dataset sizes or complexity, and if not, would the authors consider evaluating this to enhance the paper's comprehensiveness?

---

> ### Author Response · Authors · 2023-11-20
> **Reponse to Reviewer TqqW**
>
> > A more comprehensive comparative analysis with current state-of-the-art methods would elucidate the advancements BOSS provides, especially in efficiency and quality.
>
> We have added more baseline methods and additional datasets to showcase the effectiveness of our proposed algorithm. The new results can be found in [Tables 1 & 2], together with Figure 4 (quantitatively) and Figure 5 (qualitatively) of our paper.
>
> > It lacks a clear comparison of resource requirements such as memory usage, power consumption, or processing time, essential for evaluating BOSS's practicality for users with limited computational resources.
>
> We have added a paragraph inside the main text of our paper focusing on this point. Please find below a copy of it.
>
> The pre-trained models are finetuned by three methods including Uniform-Reflow, Distill-k Reflow, and BOSS  in 12000 iterations. An iteration is the passing and backpropagation process for a batch including 15 samples. Due to the similar cost of training between finetuning methods, we report the average GPU hours consumed on each pre-trained model up to 12000 iterations, using NVIDIA RTX A5000.
> * CIFAR10: 3.56 training hours.
> * CelebA-HQ: 10.35 training hours.
> * LSUN-CHURCH: 13.43 training hours.
> * LSUN-BEDROOM: 14.23 training hours.
> * AFHQ-CAT: 9.30 training hours.
>
> With this limited budget of resources, the proposed method BOSS achieves significantly better performance compared to other methods in terms of FID score.
>
> > Despite the method's enhancements over existing approaches, the computational intensity of the dynamic programming algorithm and the network retraining requirement could limit its utility in resource-constrained settings.
>
> We have added a section in the Appendix (Section 3) with a pseudocode that provides details on our Dynamic Programming algorithm. The time complexity of this algorithm is $O((K^{max})^2 \times K)$. The choice of $K^{max}$ is crucial, aiming for the Euler sampling method with $K^{max}$ stepsizes to precisely replicate the trajectory of the Ordinary Differential Equation (ODE). Typically, $K^{max}$ falls within the range of 100 to 1000, ensuring accuracy. Given this range for $K^{max}$ and the variable $K$ ranging from 2 to 1000, the algorithm executes within seconds in all scenarios. Moreover, our solved problem is fast sampling in flow-matching models which focuses on relatively small values of K (<20), so the intensity of the dynamic programming algorithm is not a worrying problem even in the case that $K^{max} > 1000$.
>
> > The scalability of BOSS, with respect to increasing dataset sizes or complexity, is also not addressed, and an evaluation of this aspect would greatly benefit the paper's comprehensiveness. How does the computational intensity of the dynamic programming algorithm and the network retraining requirement impact the method's applicability in resource-constrained environments?
>
> We agree that this is one limitation of the paper since we have not tested the method on very high-resolution images like CelebA 1024x1024 and above. However, the results on 256x256 datasets (CelebA-HQ, LSUN Church/Bedroom, AFHQ) suggested that BOSS will receive more benefit with higher dimension datasets since one NFE evaluation for each of the score networks for such dataset will take more time (as we require more network parameters to generate high fidelity high-resolution images).
>
>
> > Has the scalability of BOSS been assessed in relation to increasing dataset sizes or complexity, and if not, would the authors consider evaluating this to enhance the paper's comprehensiveness?
>
> We are unsure if the reviewer means the dataset’s number of samples or the dataset’s resolution. For the former, we always fix the retraining/redress batch size at 15 samples. For the latter, at this moment we cannot find a pretrained velocity network for higher resolution images more than 256x256. We will be thankful if the reviewer can give a pointer to us and we are willing to run the additional benchmark should the time allow.

---

> > ### Author Response · Authors · 2023-11-23
> > **A reminder**
> >
> > Dear reviewer,
> >
> > It is now only several hours until the author-reviewer discussion period ends. It would be really great if you can acknowledge that you have read the rebuttal and our revised version of the work. If you have any additional questions or concerns, we will try our best to address them quickly. Thank you for your valuable feedback, and we sincerely anticipate your response.
> >
> > Best regards,
> > The authors.

---

### Author Response · Authors · 2023-11-20
**A general response to reviewers**

We thank the reviewers for taking their time to provide us with valuable feedback. We have incorporated these suggestions into the modified version of our submission, with added elaborations and new results/plots, with additional datasets. Please refer to the updated pdf of the submission file. These new results, in additional with existing ones, show that within limited budget of resources, the proposed method BOSS achieves significantly better performance compared to other methods in terms of FID score.

Below are our general replies to the general concerns from the reviewers. We also make specific replies to each reviewer’s comment section.

> Need more comprehensive benchmarks

* We have added LSUN bedroom dataset + modified Tables 1 and 2 (FID for 4-6-8 NFEs) for redress/reflow for the CIFAR/CelebA/Church/Bedroom/AFHQ dataset.
*  We have added plots (Figure 4) demonstrating the decrease of FID with an increasing number of function evaluations (NFEs) across different sampling schemes and datasets.
* We have added qualitative comparisons for different datasets (celebA, LSUN, AFHQ) on page 9 (Figure 5) and in the Appendix.

> Resource requirements/memory usage/power consumption/processing time of the proposed method.

* We have added in Appendix A the average GPU hours, estimated on training using NVIDIA RTX A5000.

> More elaboration on the Dynamic Programming

* We have added Appendix B that provides a pseudocode and details on our Dynamic Programming algorithm.

We also apologize for replying late into the rebuttal period. We still hope that the reviewers can take their time to review our replies, and we are eager to address any further concerns or questions from the reviewers.

---

### Meta-Review · Area_Chair_NEBb · 2023-12-12

**Metareview:**

This paper utilizes dynamic programming for optimizing the step sizes to sample a pre-trained flow model. As a second step it refines the pre-trained model's weights to further improve the straightness of the sampling paths. The methods offers an improved image quality at an improved computational cost and a novel optimization for step-sizes using dynamic programming. During rebuttal the authors addressed the issues of missing comparisons to performant baseline sampling methods and missing resource/running time analysis. The reviewers felt the paper offers a sufficient contribution to justify acceptance of the paper.

**Justification For Why Not Higher Score:**

The sampling efficiency improvement offered by this method is still somewhat limited compared to the effort in designing and optimizing the solver. This is true especially in view of non-model-dependent methods used for efficient sampling of flow models.

**Justification For Why Not Lower Score:**

The paper introduces a novel step-size optimization for solvers of generative flow models which can be further developed and incorporated within other/future solver methods.

---

### Decision · Program_Chairs · 2024-01-16

Accept (poster)